# The Succinate Receptor SUCNR1 Resides at the Endoplasmic Reticulum and Relocates to the Plasma Membrane in Hypoxic Conditions

**DOI:** 10.3390/cells11142185

**Published:** 2022-07-13

**Authors:** Melanie Sanchez, David Hamel, Emmanuel Bajon, François Duhamel, Vikrant K. Bhosle, Tang Zhu, Jose Carlos Rivera, Rabah Dabouz, Mathieu Nadeau-Vallée, Nicholas Sitaras, David-Étienne Tremblay, Samy Omri, Tiffany Habelrih, Raphael Rouget, Xin Hou, Fernand Gobeil, Jean-Sébastien Joyal, Przemyslaw Sapieha, Grant Mitchell, Alfredo Ribeiro-Da-Silva, Mohammad Ali Mohammad Nezhady, Sylvain Chemtob

**Affiliations:** 1Department of Pharmacology and Therapeutics, McGill University, Montréal, QC H3A 1A3, Canada; melanie.sanchez@gmail.com (M.S.); vikrant.bhosle@mail.mcgill.ca (V.K.B.); rabah.dabouz@mail.mcgill.ca (R.D.); rouget8854@gmail.com (R.R.); alfredo.ribeirodasilva@mcgill.ca (A.R.-D.-S.); 2Department of Pharmacology, Université de Montréal, Montréal, QC H3C 3J7, Canada; david.hamel.1@gmail.com (D.H.); f.duhamel@umontreal.ca (F.D.); mathieu.nadeau-vallee@umontreal.ca (M.N.-V.); nicholas.sitaras@gmail.com (N.S.); david-etienne.tremblay@umontreal.ca (D.-É.T.); tiffany.habelrih@umontreal.ca (T.H.); 3Department of Pediatrics, Research Center-CHU Ste-Justine, Montréal, QC H3T 1C5, Canada; emmanuel.bajon@umontreal.ca (E.B.); tzhu3671@gmail.com (T.Z.); xin.hou@recherche-ste-justine.qc.ca (X.H.); js.joyal@umontreal.ca (J.-S.J.); grant.mitchell@recherche-ste-justine.qc.ca (G.M.); 4Cell Biology Program, The Hospital for Sick Children Research Institute, Toronto, ON M5G 1X8, Canada; 5Department of Ophthalmology, Research Center of Hôpital Maisonneuve-Rosemont, Université de Montréal, Montréal, QC H1T 2M4, Canada; jokarilo@gmail.com (J.C.R.); samy.omri@gmail.com (S.O.); mike.sapieha@umontreal.ca (P.S.); 6Department of Pharmacology, Université de Sherbrooke, Sherbrooke, QC J1H 5N4, Canada; fernand.junior.gobeil@usherbrooke.ca; 7Program of Molecular Biology, Faculty of Medicine, Université de Montréal, Montréal, QC H3T 1J4, Canada; 8Research Center-CHU Ste-Justine, Departments of Pediatrics, Ophthalmology, and Pharmacology, Faculty of Medicine, Université de Montréal, 3175, Chemin Côte Ste-Catherine, Montréal, QC H3T 1C5, Canada

**Keywords:** SUCNR1/GPR91, succinate, GPCR, N-glycosylation, endoplasmic reticulum, angiogenesis

## Abstract

The GPCR SUCNR1/GPR91 exerts proangiogenesis upon stimulation with the Krebs cycle metabolite succinate. GPCR signaling depends on the surrounding environment and intracellular localization through location bias. Here, we show by microscopy and by cell fractionation that in neurons, SUCNR1 resides at the endoplasmic reticulum (ER), while being fully functional, as shown by calcium release and the induction of the expression of the proangiogenic gene for VEGFA. ER localization was found to depend upon N-glycosylation, particularly at position N8; the nonglycosylated mutant receptor localizes at the plasma membrane shuttled by RAB11. This SUCNR1 glycosylation is physiologically regulated, so that during hypoxic conditions, SUCNR1 is deglycosylated and relocates to the plasma membrane. Downstream signal transduction of SUCNR1 was found to activate the prostaglandin synthesis pathway through direct interaction with COX-2 at the ER; pharmacologic antagonism of the PGE_2_ EP_4_ receptor (localized at the nucleus) was found to prevent VEGFA expression. Concordantly, restoring the expression of SUCNR1 in the retina of SUCNR1-null mice renormalized vascularization; this effect is markedly diminished after transfection of the plasma membrane-localized SUCNR1 N8A mutant, emphasizing that ER localization of the succinate receptor is necessary for proper vascularization. These findings uncover an unprecedented physiologic process where GPCR resides at the ER for signaling function.

## 1. Introduction

Succinate has long been recognized as a Krebs cycle intermediate in the mitochondria, where it is oxidized by succinate dehydrogenase, providing electrons to the electron transport chain [1,2]. Although this role is essential to life, succinate and its precursor succinyl-CoA are also byproducts of several other cellular reactions such as HIF1a stabilization and DNA methylation [3,4]. In addition, succinylation is a potent post-translational modification, although the precise mechanism and effects of such modifications need further elucidation [5,6]. Cellular succinate levels thus need precise regulation to avoid uncontrolled downstream effects [7,8,9]. Notably, circulating succinate increases in pathological conditions such as diabetes, ischemia, and rheumatoid arthritis; it is also elevated in blood samples from patients with traumatic injuries [10,11,12,13,14,15,16]. Succinate levels are increased in hypoxic conditions and under increased metabolic demand, which induces a switch from oxidative phosphorylation to anaerobic glycolysis, as was shown in vivo and in vitro [14,17,18,19,20,21]. Thus, it is essential to sense succinate levels to prevent the undesirable and potentially deleterious effects of its accumulation.

The succinate receptor (SUCNR1) is a G-protein-coupled receptor (GPCR) specific to succinate. It has an EC_50_ between 26 and 79 µM, which corresponds to pathophysiologic succinate concentrations [22,23]. SUCNR1 is coupled to Gi and Gq proteins, which lead to decreased protein kinase A (PKA) activity and induce intracellular calcium release, respectively [22,23,24]. Stimulation of SUCNR1 activates the renin–angiotensin pathway via prostaglandins synthesis [19,22,25,26]. In macrophages. stimulation of the succinate receptor increases the inflammatory response mediated by Toll-like receptor (TLR) [15,27]. Importantly, previous studies from our laboratory established the angiogenic properties of the succinate/SUCNR1 axis in neurons, introducing the notion that succinate can link metabolic demand to vascular supply, notably by inducing proangiogenic and inflammatory responses [14,19,28]. Although several recent pharmacological studies refined our understanding of the mechanism of SUCNR1 activation, insight into its regulation and its downstream cellular partners leading to its broader physiological role is lacking. Notably, in the retina (and brain), SUCNR1 is primarily expressed in the cell body, unlike in HEK 293T and Canine kidneys cells, where it appears to be located at the plasma membrane [14,22,26,29].

The concept of an extracellular ligand activating a plasma membrane 83 (PM)-bound receptor has, for a long time, been the dogma in cell biology. However, examples abound of intracellular transmembrane receptors that signal via mechanisms that are analogous to, yet different from those of their cell-surface counterparts. Nuclear localization has been reported for dozens of GPCR; such localization is either as a steady-state compartmentalization, or induced upon stimulation of receptors originally located at the PM [30,31]. Importantly, for a given receptor, nuclear and PM localizations evoke different transcriptional responses, a mechanism notably demonstrated for protease activated receptor 2, the platelet-activating factor receptor, and metabotropic glutamate receptor 5 [32,33,34]. This differential response from a receptor depending on the subcellular context at a given time, also called location-biased signaling, is increasingly recognized, and is thought to play an essential role in proper signal transduction [31,35].

Here, we show that SUCNR1 resides at the endoplasmic reticulum (ER), and we unveil the regulatory mechanism in which glycosylation is essential for this intracellular location. Intriguingly, a decrease in oxygen levels alters SUCNR1 glycosylation and thus its subcellular location, in turn altering downstream gene expression.

## 2. Material and Methods

### 2.1. Cells Culture and DNA Transfection

In vitro experiments were performed in retinal ganglion cells (RGC-5) or human embryonic kidney cells (HEK 293T). RGC-5 cells were a gift from Neeraj Agarwal (University of North Texas Health Science Center, Fort Worth, TX, USA). Typically, 50–60% confluent cells were transfected using the plasmid vector pcDNA 3.1 (Invitrogen, Carlsbad, CA, USA) coding for the different constructions of SUCNR1. Point mutations of SUCNR1 were generated by PCR mutagenesis using *Pfu* polymerase (Agilent, Santa Clara, CA, USA). All constructs were verified by automated sequencing. For each well, 2 μg of plasmid DNA were incubated with the transfection agent PEI (Polysciences #23966), and were consequently delivered on cells following a 30 min incubation period at RT. Cells were cultured in Dulbecco’s modified Eagle’s medium (DMEM) enriched with 10% fetal bovine serum, and incubated at 37 °C in a humidified atmosphere with 5% CO_2_.

### 2.2. Chemicals

Chemicals purchased were: succinate (Sigma Saint Louis, MO, USA; #S3674), L-161,982 (Cayman Chemical, Ann Arbor, MI, USA; #10011565), and prostaglandin E2 (Cayman Chemical; #14010).

### 2.3. Immunohistochemistry

Localization of endogenous SUCNR1 was determined by fluorescence microscopy using anti-SUCNR1 (Novus Biologicals, Centennial, CO, USA; #NBP-00861, 1:300). Colocalization was achieved with organelle-specific markers such as BiP (Abcam, Cambridge, MA, USA; 1:300), KDEL receptor (Abcam; 1:300), and calnexin (Millipore #MAB31261:300). Briefly, 30–40% confluent RGC-5 cells were cultured on cover slips and fixed in 2% paraformaldehyde. Next, cells were permeabilized or not in 0.1% Triton X-100 diluted in PBS and blocked with 5% goat serum. Cover slips were incubated overnight with an appropriate combination of primary antibodies. Subsequently, suitable secondary antibodies conjugated to ); Alexa Fluor 488 (Invitrogen #A11070), conc. 1:1000; and Alexa Fluor 594 (Invitrogen #A11012), conc. 1:1000 were prepared and delivered on cover slips for 1 h at RT; washes were carried out with PBS. Nuclei were stained with DAPI (Invitrogen; 1:5000). Images were taken with a laser scanning confocal microscope (Olympus FV1000, Olympus Corp., Tokyo, Japan). In transfected cells, HEK 293T cells were seeded onto a 6-well culture dish containing coverslips, and transfected with the appropriate plasmid, SUCNR1, SUCNR1 N8A, or SUCNR1 N168A, as required. The final confluency rate was kept below 30%. N-terminal FLAG-tagged SUCNR1 was visualized as described above using a FLAG-specific antibody (Sigma; 1/500), and was co-stained with either ER markers calnexin or pan-Cadherin (anti-pan-Cadherin; Cell Signaling Technology [CST], Danvers, MA, USA; 1:300).

### 2.4. Electron Microscopy

Specimens for electron microscopy were prepared as previously described [36]. Briefly, 50 μm vibratome sections of cortex from male Sprague-Dawley rats were incubated with rabbit anti-SUCNR1 antibody (1:50) overnight at 4 °C, followed by another overnight incubation with goat antirabbit gold (10 nm)-conjugated IgG (1:50) (British Biocell International, Cardiff, UK). Specimens were then postfixed in 1% osmium tetroxide, dehydrated in graded ethanol, and embedded in Epon according to the standard technique. Ultrathin sections were cut using a Reichert Ultracut ultramicrotome (Reichert-Jung, Vienna, Austria), mounted on Formvar-coated copper grids, stained with uranyl acetate and lead nitrate, and examined with a Philips 410LS transmission electron microscope (Philips, Amsterdam, The Netherlands).

### 2.5. Fraction Purification/Preparation

Subcellular fractions of RGC-5 cells were obtained as previously described [36]. Briefly, cells were resuspended in ice-cold HEPES buffer using a prechilled Dounce homogenizer for 40 min on ice. Fractions were obtained by discontinuous sucrose gradient ultracentrifugation. Briefly, crude nuclear fractions were resuspended in buffer containing 1.2 M sucrose and layered on top of a buffer solution containing 1.8 M sucrose prior to centrifugation at 60,000× *g* for 60 min at 4 °C. The purity of each subcellular fraction was validated by assessing the presence of Na^+^/K^+^ ATPase, calnexin, and lamin A/C proteins. Reactions were subsequently performed for 30 min at 37 °C prior to extracting RNA from nuclei for gene expression analysis.

### 2.6. Gene Expression Quantification

RGC-5 or HEK 293T cells were seeded in 6-well plates and treated the next day or 36 h post transfection with or without 100 μM succinate for 4 h in serum-depleted DMEM. Cells were rapidly preserved in TRIzol (Invitrogen). RNA was extracted according to the manufacturer instructions, and cDNA was synthesized using qScript cDNA SuperMix (Quanta Biosciences, Beverly, MA, USA). Primers were designed using NCBI Primer Blast. Quantitative gene expression analysis was performed on an MX3000P (Agilent) with SYBR Green Master Mix (BioRad, Hercules, CA, USA). Expression was normalized to 18S universal primer (Invitrogen; #AM1718). Expression of SUCNR1 in HEK 293T cells was determined by PCR using TAQ polymerase (Thermo Scientific #EP0405).

### 2.7. Calcium Release Kinetic

Cellular Ca^2+^ signals were measured using the fura-2-AM technique, as previously described [36]. Briefly, RGC-5 or shGFP/shSUCNR1 stably expressing RGC-5 cells were loaded with fura-2-AM and stimulated with 100 μM of succinate. In another experiment, RGC-5 isolated nuclei or nuclei+ER or whole cells were treated with 100 μM of succinate. Intracellular calcium signals were measured by spectrofluorometry (LS50B, PerkinElmer, Beaconsfield, UK) and the fluorescent signal appropriately calibrated. The calcium concentrations were calculated according to Grynkiewicz et al. [37].

### 2.8. Western Blot

To determine protein levels, Western blot analysis was carried out in RGC-5 cells or by transfecting 293T cells with myc-tagged SUCNR1 or mutants SUCNR1 N8A or SUCNR1 N168A. ERK and AKT phosphorylation were examined after treatment with 100 μM succinic acid in serum-starved DMEM at several time points. Cells were harvested using RIPA buffer. Immunoblotting was performed using specific antibodies: anti-SUCNR1 (Novus Biologicals; #NBP1-00861), Myc-tag (CST; #2276), p42/p44 (CST; #4376S), and phospho-p42/p44 (CST; # 4695S). Signals were revealed by chemiluminescence using appropriate horseradish peroxidase-conjugated secondary antibodies, and observed with an Image Quant LAS 500 (GE healthcare, Boston, MA, USA). Total protein content was normalized using an anti-β-actin antibody (Santa Cruz Biotechnology, Dallas, TX, USA, [SCBT]) where appropriate.

For ERK or AKT activation, the membranes were stripped and reblotted with an antibody against total ERK or AKT. The activation was estimated by comparing the signal between the phosphorylated and total forms of ERK/AKT using ImageJ (NIH, Bethesda, MD, USA) from at least three independent experiments. For Western blot analysis, cells were treated as described above.

### 2.9. Retinal Flatmounts

Eyes were enucleated and fixed in 4% paraformaldehyde (PFA) for 90 min at 4 °C, and then stored in PBS at 4 °C. For each eye, the cornea and lens were removed, and the retina was gently separated from the underlying choroid and sclera under a dissecting microscope. Subsequently, retinas were incubated for 10 min in 100% cold methanol (−20 °C), then incubated overnight at 4 °C in 1% Triton X-100/1 mM CaCl_2_/1× PBS with the TRITC-conjugated lectin endothelial cell marker *Bandeiraea simplicifolia* (1:100; Sigma-Aldrich, St-Louis, MO, USA). Lectin-stained retinas were whole-mounted onto Superfrost/Plus microscope slides (Thermo Fisher Scientific, Waltham, MA, USA) with the photoreceptor side down, imbedded in Fluoro-gel, and imaged at 10× using a Zeiss AxioObserver.Z1 (Zeiss, Jena, Germany). Images were merged into a single file using the MosaiX option in AxioVision software version 4.6.5 (Zeiss). The density of the capillary network was assayed by quantification of FITC-stained vessels using ImageJ, as previously reported [38].

### 2.10. Intraocular Succinate Injections and Retinal Vascular Density Quantification

Mouse pups were injected at postnatal day 4 (P4) with succinate (final concentration 100 μM) with or without probenecid (final concentration 1 mM) using a 10 μL Hamilton syringe (1.0 μL final volume of injection). Animals were sacrificed and perfused with 4% PFA at P8. The density of the capillary network was assayed by quantification of *Bandeiraea simplicifolia* (1:100, Sigma) stained flat-mounted using ImageJ, as previously reported [38].

### 2.11. Immunoprecipitation

Coimmunoprecipitation analysis was performed on lysates from cells transfected with wild-type or mutated SUCNR1 (myc-tagged). Cell lysates were precleared before incubation with an anti-SUCNR1 antibody (Novus) and protein A/G-Sepharose beads at 4 °C. Precipitates were washed in lysis buffer, except the NaCl concentration was raised to 0.7 M and no SDS was added to the lysis buffer. Samples were resolved by SDS-PAGE, and analyzed by Western blotting using the indicated antibody: RAB2 (SCBT), RAB11 (SCBT), COX-2 (CST), cPLA_2_ (SCBT), and HIF-1α (Novus). Membranes were also probed with an anti-SUCNR1 antibody as a recovery control.

### 2.12. Aortic Explants

Aortae from adult Sprague-Dawley rats were cut into 1 mm thick sections. Rings embedded in growth-factor-reduced Matrigel (BD Biosciences Billerica, MA, USA; #354230) in 24-well plates were cultured for 3 days in conditioned media obtained from HEK 293T transfected with either SUCNR1 or SUCNR1-N8A construction or left untransfected and stimulated with 100 μM succinate or not for 24 h. Photomicrographs of individual explants were taken each day, and vascular sprouting was measured by the area covered by the outgrowth of the aortic ring using ImageJ.

### 2.13. PGE2 Dosage

PGE_2_ was extracted with SPE purification tC18 columns (Waters, Milford, MA, USA) from cultured media of RGC-5 stimulated or not with 100 μM succinate. Quantification was performed with a PGE_2_ EIA Kit (Cayman Chemical). The amount of PGE_2_ was normalized to the protein content.

### 2.14. Lentivirus Production and Intravitreal Injections

Lentiviral vectors (pLenti-X1-puro) were prepared as we previously reported by transfecting HEK 293T cells with a vector plasmid containing either SUCNR1-wt or SUCNR1-N8A, together with the third-generation packaging plasmids pV-SVG, pMDL, and pREV (Open BioSytems, Huntsville, AL, USA). Secreted viruses were filtered (0.22 µm) and ultracentrifuged at 50,000× *g* (2 h, 10 °C; L8-70M, Beckman). The viruses were then resuspended in PBS, aliquoted with consistent amounts of particles, and stored at −80 °C.

### 2.15. Intraocular Lentiviral infection of SUCNR1-wt and SUCNR1-N8A in Mice

P2 mouse pups were anesthetized with 3.0% isoflurane in oxygen and injected intravitreally with 1.0 µL of LV particles containing either SUCNR1-wt or SUCNR1-N8A using a 10 µL Hamilton syringe fitted with a 50-gauge glass capillary tip. Animals were sacrificed at P7, and vascular surface area was measured on retinal flatmounts.

### 2.16. Statistical Analysis

We performed between-group comparisons using one-way analysis of variance, followed by Bonferroni’s test to compare means. Data are presented as means ± s.e.m.

## 3. Results

### 3.1. SUCNR1 Is Located at the Endoplasmic Reticulum

Prior immunofluorescence (IF) observations from our laboratory and others have revealed that the succinate receptor localizes throughout the whole cell body in different organisms, tissues, and cell types [14,28,39]. To refine the subcellular location of the receptor, we performed colocalization analyses with known intracellular markers, relying on mouse retinal ganglion neurons as a model. The SUCNR1 IF signal strikingly colocalized with that of ER-specific markers KDEL and GRP78/BiP (Figure 1A). To ascertain this intracellular staining pattern of SUCNR1 in neurons, we performed in vivo investigations through high-resolution immunogold electron microscopy on rat cortical brain sections. Confirming the aforementioned results, SUCNR1 expression was largely confined to the ER at the ultrastructural level (Figure 1(BI,BII)); essentially, no signal could be detected at the nuclear envelope, which is in continuum with the ER [40] (Figure 1(BI,BII)). Although SUCNR1 immunogold staining was found at the plasma membrane, this was considerably less than at the ER (Figure 1(BII)). The immunogold signal was also detected in the vicinity of mitochondria (Figure 1(BIII,BIV)), which physically interacts with the ER. Of note, succinate treatment did not affect this intracellular localization (Appendix A).

The presence of GPCR at the ER could be attributed to specific motifs that affect protein folding or transport. To confirm that SUCNR1 location to the ER is not merely due to a synthesis process, we investigated whether this localization is associated with a functional role in a cell-free system. To assess this, we relied on subcellular fractionation procedures yielding either cell nuclei with a preserved ER network (Nucl + ER) or nuclei devoid of ER (Nucl) (Appendix A) [36]. As stimulation of SUCNR1 with succinate was previously shown to trigger the expression of proangiogenic genes, we used *Vegfa* expression as a readout of functional SUCNR1 signaling [14]. Upon stimulation of these cell-free systems, we found increased *Vegfa* expression from whole-cell extracts (WC) or in nuclei in the presence of the ER (Figure 1C). Remarkably, the nuclear fraction alone, devoid of SUCNR1, did not lead to increased expression of *Vegfa* (Figure 1C). We also verified that calcium release, another known feature of SUCNR1 activation [22,24], was effectively triggered by the ER-resident receptor. Relying on the same ER-containing cellular fractions, we measured Ca^2+^ release using the ratiometric fluorescent dye FURA-2 (Figure 1D and Appendix A). Compared with the calcium released upon treatment with the calcium ionophore ionomycin, stimulation of WC and Nucl + ER with succinate triggered a notable and expected calcium release (Figure 1D). In contrast, isolated nuclei devoid of ER showed little release of the calcium stored in their envelope compared with ionomycin (Figure 1D). To confirm our observations that succinate treatment, leading to calcium release and *Vegfa* gene expression, are mediated by SUCNR1, we knocked down the expression of the receptor in RGC cells, which notably impaired ERK phosphorylation upon succinate stimulation (Appendix A). Succinate treatment of the Nucl + ER fractions from SUCNR1 knocked-down cells showed blunted *Vegfa* expression and reduced calcium release (Figure 1E,F and Appendix A). These results suggest that ER-resident SUCNR1 is functional and fully able to signal upon succinate stimulation.

### 3.2. SUCNR1 Motifs Responsible for ER Localization: Glycosylation

To elucidate the potential mechanisms that drive the retention of a functional succinate receptor at the ER, we examined the amino acid sequences of mouse, rat, and human SUCNR1 for clues of intracellular retention signals. No consensus regarding ER retention signals was found in SUCNR1 (Appendix A), suggesting that ER localization is likely dependent on alternative mechanisms.

The SUCNR1 receptor contains two phylogenetically conserved N-glycosylation (NxS/T) consensus sequences in extracellular domains [26] (Appendix A). Post-translational glycosylation regulates protein subcellular localization, usually by targeting transmembrane proteins to the plasma membrane [41,42,43]. Glycosylation sites were found to be located on the N-terminal extracellular tail and the second extracellular loop, at positions N8 and N168 of SUCNR1, respectively (Figure 2A). To investigate the role of these glycosylation sites in SUCNR1 localization, we generated N-terminally FLAG-tagged SUCNR1 expression vectors carrying missense mutations for either site, substituting glycosylation-incompetent alanine for the wild-type asparagine (N8A and N168A); these plasmids were used to transfect HEK 293T cells. By SDS-PAGE, the overexpressed wild-type SUCNR1 yielded a signal appearing as a smear with a higher molecular weight (MW) than was predicted by the sole protein sequence (Figure 2B). As expected for glycosylation-deficient mutants, both N8A and N168A proteins migrated at a lower molecular weight than the wt SUCNR1. Yet, mutant receptors migrated to approximately 40 to 45 kDa, higher than expected for unmodified SUCNR1, suggesting that glycosylation-deficient mutants were still post-translationally modified (Figure 2B). To confirm that this remarkable migration pattern is due to the receptor being glycosylated, cell extracts were treated with peptide-N-glycosidase F (PNGase F), which cleaves off glycosyl residues from Asn. As expected, SUCNR1 from PNGase-treated extracts migrated to its theoretical size of 37 KDa (Figure 2C). Of note, both N8A and N168A receptors also migrated to their theoretical size after PNGase treatment, suggesting that these mutations do not affect glycosylation of the other consensus N-glycosylation site (Figure 2C and Appendix A).

We proceeded to examine if glycosylation of SUCNR1 can regulate its subcellular location. In SUCNR1-expressing HEK 293T cells, the IF signal revealed the wt SUCNR1 at its expected perinuclear localization, colocalizing with the ER marker calnexin (Figure 2D). Under the same conditions, the N8A mutant showed an IF signal exclusive of the calnexin marker, suggesting a mislocalization of the mutant SUCNR1 (Figure 2D); in contrast, the N168A mutant receptor is ER-localized (Figure 2D), inferring the critical role of N8 glycosylation in this process. In other cell types, such as MDCK cells, SUCNR1 is expressed on the PM; this process often involves receptor glycosylation. To ascertain whether the glycosylation status of SUCNR1 plays a role in localization, we performed IF on nonpermeabilized cells, which only allowed immuno-detection of PM-localized proteins. In these conditions, FLAG-tagged native SUCNR1 failed to be detected, consistent with its intracellular location (Appendix A and Figure 2D). However, the PM surface localized FLAG-tagged N8A mutant was detected in nonpermeabilized cells, and consequently colocalized specifically with the PM marker cadherin (Appendix A, right panel). In contrast, the FLAG-tagged N168A mutant receptor was undetectable in nonpermeabilized cells (Appendix A). Overall, these results suggest a mechanism where SUCNR1 Asn 8 and 168 are both independently glycosylated, whereas only the former is required for the ER retention of the receptor.

To assess the molecular partners involved in the trafficking of SUCNR1 and mutants, we focused on rab GTPases, which are responsible for the proper shuttling of proteins between organelles, vesicle budding, and uncoating [33,44]. In this context, RAB2 is specifically involved in shuttling from the Golgi to the ER and RAB11 in delivering proteins to the PM [45,46,47]. Co-immunoprecipitation (coIP) experiments revealed that wild-type SUCNR1 and SUCNR1 N168A interact with RAB2 but not with RAB11, whereas SUCNR1 N8A interacts with RAB11 and interacts little with RAB2 (Figure 2E). This suggests that the glycosylation state of SUCNR1 determines its interaction with shuttling partners, thus affecting its subcellular location.

### 3.3. Subcellular Localization of SUCNR1 Affects Downstream Signaling

For a given receptor, signaling and function differ depending on its cellular location. We thus determined if this principle also applies to SUCNR1, and investigated whether the N8A mutant located at the cell membrane triggers downstream signaling pathways differently than the ER-resident receptor. Notably, SUCNR1 stimulation with succinate has been reported to activate the MAPK pathway and pro-survival signals in kidney and neurons [19,22,25]. We assessed ERK and AKT phosphorylation upon stimulation of wild-type and mutant SUCNR1 isoforms expressed in HEK 293T cells. The wild-type and N168A receptors, but not the PM-localized N8A mutant, elicited prolonged AKT activation (Figure 3A). However, ERK phosphorylation was triggered to similar levels with all three isoforms of the receptor (Figure 3B). We then examined if gene expression differs according to SUCNR1 subcellular location and altered signaling. We measured the expression of angiogenic genes previously reported to be regulated by SUCNR1 [14,39]. Succinate stimulation of the wild-type SUCNR1 in HEK 293T cells, but not of the PM-localized N8A mutant, induced strong *VEGFA* expression (Figure 3C). Concordantly, intravitreal injection of succinate at P4 in mice triggered increased vascular density, which was prevented upon organic anion transporter inhibition by probenecid, suggesting that succinate requires access to its intracellular receptor site (Appendix A). Of note, in cultured RGC-5 cells, the succinate-driven *VEGFA* expression was reversed by probenecid (Appendix A). In contrast, succinate stimulation of PM-localized SUCNR1 N8A elicited a more robust induction of *FGF2*, *ANGPT1*, and *ANGPT2* than in cells expressing the ER-localized wild-type receptor (Figure 3D–F). These results point to a location bias of SUCNR1 depending on its glycosylation state, where an altered glycosylated receptor localizes to the PM and triggers different signaling pathways, which, in turn, lead to different gene expressions.

### 3.4. SUCNR1 Glycosylation and Localization Are Regulated during Hypoxia

Glycosylation is a major post-translational modification that impacts receptor function, and is thus tightly regulated. Pathophysiologic conditions such as hypoxia alter glucose metabolic pathways and modify glycosylation processes [48]. For instance, in an anaerobic tumor environment, specific glycosylation patterns favor angiogenesis [49]. Accordingly, we evaluated whether hypoxia affects the glycosylation status of SUCNR1 and its cellular localization. We first confirmed that SUCNR1 is glycosylated in retinal ganglion cells; accordingly, the treatment of RGC extracts with PNGase F generated a 37 KDa SUCNR1 signal (Appendix A). Exposing the RGC to hypoxia (2% O_2_) led to the gradual appearance of a lower molecular weight, while the higher-molecular-weight SUCNR1 concomitantly disappeared (Figure 4A). This suggests that the receptor is differentially N-glycosylated upon hypoxic challenge. We then assessed whether localization of SUCNR1 is affected by hypoxia-related post-translational modifications. Under hypoxic conditions, SUCNR1 (detected by IF) relocated time-dependently from the ER to a more diffuse localization, largely at the PM (Figure 4B). Consistent with the purported role of RAB11 in SUCNR1 localization, suppression of RAB11 in RGC retained SUCNR1 at the ER colocalized with calnexin during hypoxia (Figure 4C). Furthermore, exposure of cells to hypoxia led to increased expression of three genes that were upregulated by the PM-localized N8A SUCNR1 (i.e., *Angpt1*, *Angpt2*, and *Fgf2*). This increased expression of angiogenic genes was abrogated upon SUCNR1 knock down (Appendix A). This suggests that the deglycosylated SUCNR1, not contained at the ER during hypoxia, is necessary to drive the expression of these proangiogenic genes.

To confirm that the physiological regulation of the receptor by O_2_ concentration occurs in vivo, we verified the localization of SUCNR1 in the retina of mice sequentially exposed to hyperoxia followed by normoxia, which subjects the retina to a hypoxic environment, as seen in oxygen-induced retinopathy (OIR). In the intact mouse, the SUCNR1 IF signal (as seen in the ganglion cell layer) exhibits sparse colocalization with the PM-specific cadherin signal (Figure 4D). However, during the hypoxic period of OIR, SUCNR1 staining clearly colocalizes with cadherin (Figure 4D). These in vivo results reproduced our observations in cultured cells (Figure 4C), and indicate that SUCNR1 glycosylation and, in turn, subcellular localization is physiologically regulated.

### 3.5. Mechanism of Gene Induction by ER-Localized SUCNR1

Succinate-receptor-triggered *VEGFA* expression occurs via the COX-2/PGE_2_/EP_4_ axis [19,25]. We explored if this effect is specifically mediated by ER-resident SUCNR1. The results of our immunofluorescence experiments revealed that cPLA_2_ and COX-2 colocalized with calnexin in RGC (Appendix A). The interaction of these enzymes with SUCNR1 was biochemically confirmed by co-immunoprecipitation (co-IP) on whole cell extracts and microsomal fractions of RGC (Figure 5A). The succinate stimulation of cells increased the interaction between COX-2 and, to a lesser extent, cPLA_2_ with SUCNR1; the rate-limiting enzyme COX-2 specifically interacted with ER-localized wild-type SUCNR1 (and N168A mutant) in HEK 293T cells, but not with PM-localized N8A mutant (Figure 5B). Accordingly, succinate treatment increased PGE_2_ generation in whole RGC as well as in the ER fractions of these cells (Appendix A and Figure 5C), clarifying that SUCNR1 at the ER compartment displays the enzyme machinery to trigger prostaglandin synthesis. Consistent with PGE_2_ receptor (EP_1_, EP_3_, and EP_4_) localization at the nuclear envelope where they trigger gene transcription [50,51,52], we observed that ER pretreated with succinate and subsequently combined with the cell nuclear fraction led to *Vegfa* expression, which was abolished by EP_4_ -specific antagonist L-161,982 (Figure 5D). Overall, this suggests that glycosylation of SUCNR1, which permits retention of the receptor at the ER, is essential for proper interaction with the PG synthesis pathway, and subsequent activation of the nucleus-localized EP_4_ receptor, to elicit *Vegfa* expression upon succinate stimulation.

### 3.6. ER Localization Is Essential for Angiogenesis Mediated by SUCNR1

Finally, to ascertain functional *VEGFA* regulation by wild-type SUCNR1, we first studied the ex vivo vascular sprouting of rat aortic explants exposed to culture media from HEK 293T cells expressing wild-type or N8A SUCNR1 mutant. Explant sprouting was significantly increased by conditioned media from SUCNR1- but not the N8A mutant-expressing HEK 293T cells stimulated with succinate (Figure 6A), consistent with the increased expression and effects of ANGPT1 without those of VEGFA [32,53]. This angiogenic effect of SUCNR1 was also tested in vivo. We noticed that *Sucnr1*^−/−^ pups presented a temporally restricted defect in retinal vascularization from P3 to P17 (Figure 6B,C). The brevity of this defect is concordant with the notion that the succinate receptor, although transiently important for normal vascularization, is not essential for the final proper development of the retina in rodents [14,28,54]. Lentivirus expressing wt or N4A versions of the mouse SUCNR1 (Appendix A) were injected in *Sucnr1*^−/−^ pups at P2, and retinas were collected at P7 to assess vascularization. Infection with the wild-type SUCNR1 effectively rescued the defective vascularization, whereas infection with the N4A mutant showed a significantly smaller vascular area (Figure 6D). Overall, glycosylated ER-localized SUCNR1 is essential for mediating the proangiogenic effect of succinate.

## 4. Discussion

ER localization is often associated with protein folding and processing, a maturation process usually not compatible with functional proteins. Yet, we hereby present unprecedented evidence that a post-translationally modified ER-resident GPCR, namely SUCNR1, elicits a potent proangiogenic response upon stimulation with succinate. Its ER localization was corroborated by microscopy and biochemical methods, and is permitted by N-glycosylation of its first extracellular domain. Concordant with previous reports, its proangiogenic activity correlates with the activation of the MAPK/ERK and PI3K/AKT signaling pathways [19,22,26,39]; the PGE_2_ pathway was also activated by succinate, as anticipated [19]. Correspondingly, the mutation of the N-glycosylation site at position 8 (human) fails to activate AKT, does not trigger *VEGFA* expression, and does not exhibit a robust proangiogenic response. Our findings also revealed that pathophysiologic conditions, notably hypoxia, mitigate the glycosylation state, in turn dampening the angiogenic response. The location of SUCNR1 at the ER, in close proximity to mitochondria where succinate is produced, allows for a rapid response to succinate imbalance, triggering downstream gene expression before other stress pathways are activated. In this context, therapeutic targeting of SUCNR1 needs to consider the subcellular localization that defines the signaling and induction of specific genes. A schematic diagram depicting the actions of SUCNR1 at the ER, through the PGE_2_ pathway, leading to *VEGFA* induction and angiogenesis, is presented in Figure 7.

SUCNR1 is considered a prominent sensor of oxygen, which reinstates its supply through angiogenesis [14,55]. In OIR, transient hyperoxia induces the vaso-obliteration of retinal vessels; upon resuming a normoxic environment, the relative hypoxic retina triggers an exaggerated vaso-proliferation. In this context, our observations that hypoxic conditions lead to a relocalization of SUCNR1 out of the ER, correlating to impaired downstream signaling and reduced angiogenesis of the PM-located N8A SUCNR1 mutant, allow us to propose a new model for the subcellular localization of SUCNR1 in OIR. After vaso-obliteration of the retinal vessels occurring during the hyperoxic phase of OIR, re-establishment of normal ambient O_2_ concentrations creates hypoxia, resulting in the PM relocation of SUCNR1. Upon the initial metabolic changes that occur when succinate rapidly increases with hypoxia, SUCNR1 at the ER is activated to induce VEGFA expression. Sustained hypoxia relocates SUCNR1 to the cell surface, from where it senses increased extracellular succinate to elicit vascular factors that regulate vessel sprouting and maturation, notably ANGPT1, ANGPT2, and, to a lesser extent, FGF2. Whereas ANGPT1 and ANGPT2 are generally thought to have antagonistic action on their cognate receptor TIE2, the effect of their mutual increased expression on the retina vasculature is not defined and context-dependent [56,57,58,59,60]. This deduction should be considered in the context that VEGFA is not homogeneously distributed between hypoxic and normoxic areas, partly due to the effects of repulsive and antiangiogenic factors such as semaphorins released from hypoxic neurons triggering nonuniform pathological neovascular tufts [32]. We speculate that an increase in both angiopoietins together with FGF2 in the avascular retina may drive the aberrant vascularization that results in pathological tufts. Together, our findings regarding SUCNR1 localization expand upon our understanding of the regulation of (retinal) neovascularization.

GPCRs are widely considered to be directed toward the PM; many have been shown to signal from intracellular locations such as the endosomes, nuclear membrane, or nucleoplasm [30,31,35]. To the best of our knowledge, localization to the ER was only proposed for two other GPCRs: GPR30 and mGluR5 [34,61]. However, in the present example applied to SUCNR1, we provide the first evidence showing a functional GPCR at the ER. A motif explaining ER localization applies to the canonical ER-retention sequence RXR in the ICL2 of human SUCNR1 (R^217^NR). This sequence can be recognized by coatomer protein I (COPI), which mediates retrograde transport from the Golgi to the ER [62,63,64]. However, the N8A SUCNR1 mutant that localizes to the PM carries an intact RNR sequence, suggesting that this recognition site is either hidden from the COPI proteins because of a specific conformational structure of the receptor, or not sufficiently robust to retain SUCNR1 to the ER upon deglycosylation, in a mechanism reminiscent of the GABA receptor heterodimerization, albeit with an unknown partner [65]. In addition, although this sequence is not present in rodents, we observed ER localization of SUCNR1 in rodents. Hence, although we cannot definitively rule out a role for the R^217^NR sequence in the ER retention in humans, we think it is unlikely to play a major role for the aforementioned reasons. Other canonical ER-retention sequences, such as KDEL, KKXX, or HIEL motifs, are not present at the C-terminus of SUCNR1, thereby implying that localization of the receptor to the ER is not dependent on a canonical pathway. Of note, di-leucine motifs known to mediate the export of GPCR to the synapse of neurons can be found within the sequence of SUCNR1 [66]. This export has been shown to be mediated by an interaction with RAB8 and RAB1 [67,68]. It is thus striking that these motifs in SUCNR1 do not target the glycosylated receptor to the PM: we speculate that the overall 3D structure of the glycosylated SUCNR1 does not allow access of the PM-targeting proteins to their cognate motifs on the receptor, or that specific interactions with yet-uncovered, glycosyl-interacting proteins hide these motifs from trafficking proteins such as RAB1, RAB8, or RAB11. In addition, cell-type-specific trafficking mechanisms are likely at play, as has been shown for other GPCRs [69,70,71,72]. Future work to uncover the motifs responsible for the subcellular sorting of SUCNR1 is warranted, and will likely open new grounds for GPCRs and other transmembrane proteins.

We noted that, in all species where an ortholog of *SUCNR1* is found, the N-terminal N-glycosylation site is conserved. This conservation underlies the importance of this post-translational modification for the proper function of the receptor. A second glycosylation site, located in the ECL2 of the human *SUCNR1* (N168), is absent in rodents. Strikingly, SUCNR1 N168A, although still observable at the ER, exhibited a different downstream gene expression than N8A, suggesting N168 glycosylation may be required for proper signaling in humans. The absence of this second glycosylation site in rodents may underlie different needs in response to succinate.

Localization of SUCNR1 in various cells has mostly been described at the PM, where it can be internalized upon succinate stimulation [22,23]. This contrasts our findings where SUCNR1 in transfected cells, as well as in intact retinal tissue, inherently localized at the ER compartment. These apparent distinctions likely result from different intracellular transport mechanisms, such as those that apply to cell-type- and cell-condition-specific RAB protein expression [73,74,75,76]. Such mechanisms may explain why proper glycosylation of SUCNR1 is associated with plasma membrane localization in MDCK cells, and with ER localization in HEK 293T and RGCs. Although glycosylation is part of the synthesis pathway of transmembrane proteins thought to be essential for proper folding and addressing of membranes after final processing at the Golgi [77,78], exceptions to this have been reported. For instance, the ER-resident chaperone GRP94 physiologically carries an N-glycosylation, yet remains at the ER; in contrast, N-glycosylation mutants of the GPCRs DOR1 and M2AchR localize to the plasma membrane [79,80,81]. Hence, glycosylation is not exclusively associated to extracellular milieu interactions, and likely depends on a cell-specific interplay between protein conformation and subcellular trafficking.

SUCNR1 couples to PGE_2_ synthesis, ensuing in *VEGFA* expression via EP_4_ receptor activity, itself localized at the cell nucleus [19,50,82]. The location of SUCNR1 at the ER is fitting, in proximity to the prostaglandin synthesis pathway, documented to be present at the ER, as is the case for COX-2, cPLA_2_, and mPGES [83,84]. Such a physical proximity between these actors may help mediate a rapid response to intracellular succinate accumulation, favoring vascularization to alleviate new cellular metabolic requirements. Along these lines, a direct interaction of COX-2 with regulatory proteins, including several GPCRs, was described [85].

The model we propose based on our observations suggests that succinate accesses the ER lumen to activate SUCNR1. Although to the best of our knowledge, no reports concerning succinate concentration in the ER have been published, inferences on this mechanism can be drawn based on published literature. Several reports have shown that the ER is highly permeable to small molecules such as α-d-glucopyranoside, biotin, and ATP [86,87,88]. While the transport mechanism of such small molecules still remains elusive, considered pathways include: passive diffusion through the more permeable, cholesterol-poor lipid bilayer of the ER; diffusion across the large translocon pore; active transport along the maturation process of transporters en route toward the plasma membrane; or through the Mitochondria-Associated Membrane (MAM) [86,88,89,90]. Either or all of these mechanisms may contribute to allowing succinate into the ER lumen to activate SUCNR1.

In conclusion, our observations unveil an unprecedented localization of a functional GPCR, SUCNR1, at the ER, which depends on N-terminal glycosylation. Deglycosylation directs the receptor to the PM and modifies its signaling. The present findings expand upon our understanding of GPCRs in other subcellular compartments, notably the nucleus, mitochondria, and endosomes [31]. Notions described herein also highlight the importance of the subcellular localization on GPCR function, as it adds a level of complexity to GPCR biology. These observations also have therapeutic implications; for example, targeting a specific receptor based on its localization impacts the procured actions. This was observed with NK_1_R, where endosomal targeting prolongs its antinociceptive effects [91]; another example applies to the kinin B_1_ receptor, which shows anticancer activity when targeted intracellularly, but not when at the cell surface [92]. This would also concern SUCNR1, where subcellular location affects function.

## Figures and Tables

**Figure 1 cells-11-02185-f001:**
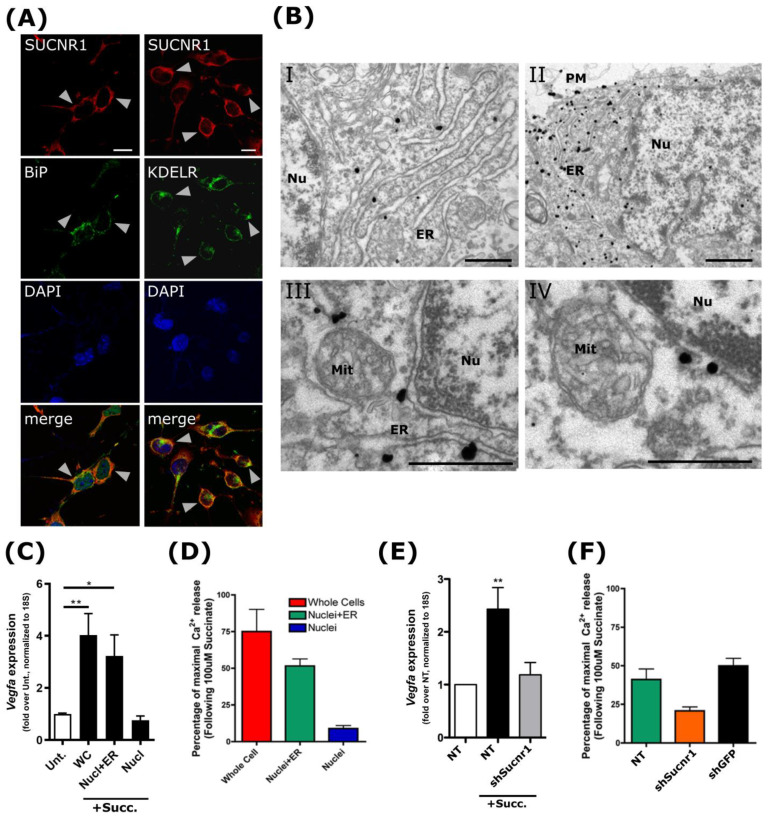
ER-localized SUCNR1 is functional. (**A**) Endogenous SUCNR1 in RGC cells colocalized with ER markers,: KDEL receptor and BiP; arrows point to the regions of highest colocalization. Scale bars: 20 μm. (**B**) In situ ultrastructural ER localization of SUCNR1 in rat parietal cortex tissue sections by electron microscopy, where immunogold-labeled antibodies against SUCNR1 could be found in ER folds. PM: plasma membrane; Nu: nuclei; ER: endoplasmic reticulum; Mit: mitochondria. Scale bars: (**I**,**II**): 1 μm; (**III**,**IV**): 400 nm. (**C**) RT-qPCR on isolated fractions of RGC-5 cells reveals that a 4 h stimulation with 100 μM succinate induced *Vegfa* expression. Absence of the ER impaired this induction upon succinate stimulation (filled bars) compared with unstimulated samples (Unt.: empty bar). WC: whole cells; Nucl + ER: fraction; Nucl: nuclear fraction only. (**D**) Ca^2+^ release on isolated RGC fractions containing ER or not. Calcium release was measured for 30 min, and is represented as fold change compared with the ionomycin-treated sample. Ca^2+^ release was not observed in absence of the ER. (**E**) Isolated Nucl + ER fractions of RGC cells expressing a shRNA against SUCNR1 did not express *Vegfa* after 100 μM succinate stimulation for 4 h. NT: nontransfected. (**F**) In isolated Nucl + ER fractions, Ca^2+^ release upon succinate treatment was lower in shSUCNR1 cells than in shGFP control cells. Calcium release was measured as in (**D**). *, **, *p* < 0.05, 0.01compared with corresponding controls, respectively.

**Figure 2 cells-11-02185-f002:**
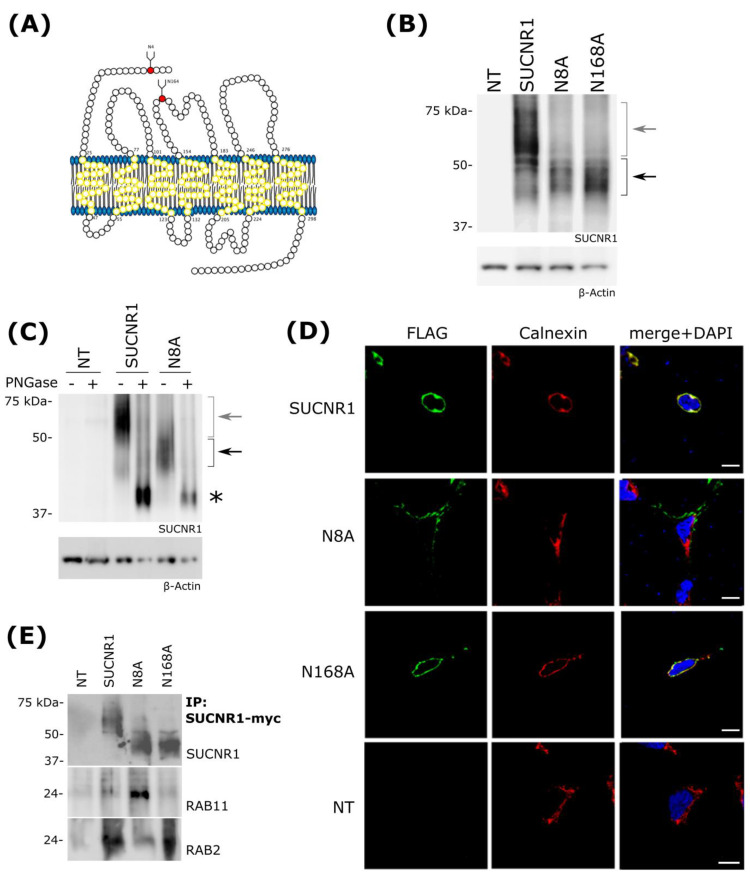
SUCNR1 N−glycosylation is necessary for ER retention. (**A**) Snake plot representation of theoretical mouse SUCNR1 topology, with potential N-glycosylation signals marked in red. (**B**) Steady-state levels of SUCNR1 and glycosylation-deficient mutants N8A and N168A in HEK 293T cells transfected with pcDNA-SUCNR1. Wild-type and mutant SUCNR1 proteins migrate as a smear characteristic of N-glycosylated proteins, albeit with a different molecular weight (wt and mutants indicated by grey and black arrows, respectively). (**C**) Transfected HEK 293T whole-cell lysates were treated or not with PNGaseF to fully deglycosylate proteins. After treatment, SUCNR1 migrated to its theoretical molecular weight of 37 KDa (*). Fully glycosylated and partially glycosylated receptors are indicated by grey and black arrows, respectively. (**D**) Cellular localization of SUCNR1 by confocal microscopy. HEK 293T cells were transfected with FLAG-tagged wild-type or mutant SUCNR1 constructs. Calnexin was used as an ER marker, and nuclei were stained with DAPI. NT: nontransfected. Scale bars: 20 μm. (**E**) Immunoprecipitation of myc-tagged SUCNR1 from HEK 293T expressing wild-type or mutant SUCNR1 constructs. SUCNR1-myc, RAB11, and RAB2 presence in the IP fraction was revealed with specific antibodies as indicated in the figure.

**Figure 3 cells-11-02185-f003:**
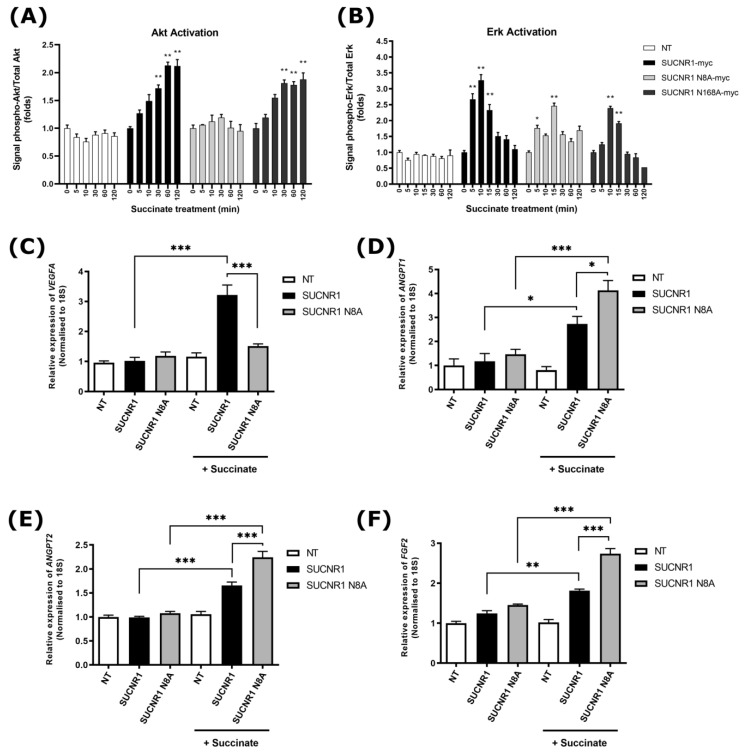
Impaired signaling downstream of N-glycosylation mutant SUCNR1. (**A**) Signal ratio of phospho- to total AKT level was measured in HEK 293T cells transiently transfected with the wild-type or glycosylation mutants SUCNR1 and treated or not with 100 μM succinate for the indicated times. NT: nontransfected. N = 4; * *p* < 0.05; ** *p* < 0.01 vs. comparative values. (**B**) Similar to (**A**), except that phosphorylation of ERK1/2 was measured. (**C**–**F**) Expression of proangiogenic genes *VEGFA*, *FGF2*, *ANGPT1*, and *ANGPT2* in HEK 293T cells transiently transfected with wild-type or mutant SUCNR1, treated or not with 100 µM succinate for 4 h. NT: nontransfected. N = 4; ** *p* < 0.01; *** *p* < 0.001 vs. comparative values.

**Figure 4 cells-11-02185-f004:**
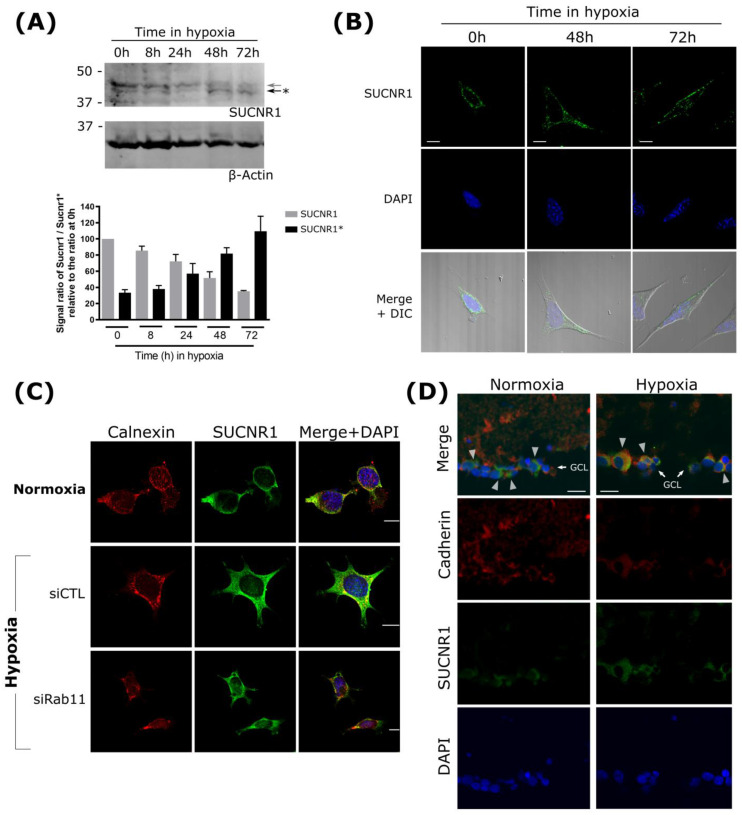
SUCNR1 N-glycosylation and relocalization during hypoxia (**A**) Western-blot analysis of endogenous SUCNR1 levels in RGC-5 cells incubated in hypoxic conditions (2% O_2_). Glycosylated and deglycosylated (*) proteins are indicated by the grey and black arrows, respectively. Actin beta was used as the loading control. Signal quantification is shown at the bottom. (**B**) RGC-5 cells were incubated in hypoxic conditions for the indicated times before fixation and immunofluorescence analysis of endogenous SUCNR1 localization. DAPI and DIC images depict the nucleus and the cell shape, respectively. Scale bars: 20 μm. (**C**) SUCNR1 immunofluorescence photomicrographs of RGC-5 cells knocked down for *Rab11* or not and incubated in hypoxic conditions or not. Calnexin signal (shown in red) delineates the ER. Scale bars: 20 μm. (**D**) SUCNR1 immunofluorescence pictures of retina cross-sections from mouse subjected to OIR (hypoxia) or normoxia show the receptor expression increases in the ganglion cell layer (GCL) in the hypoxic retina. Arrowheads on merged images indicate areas of low and high colocalization of SUCNR1 with (cell surface) cadherin, respectively, in normoxic and hypoxic conditions. Scale bars: 40 μm.

**Figure 5 cells-11-02185-f005:**
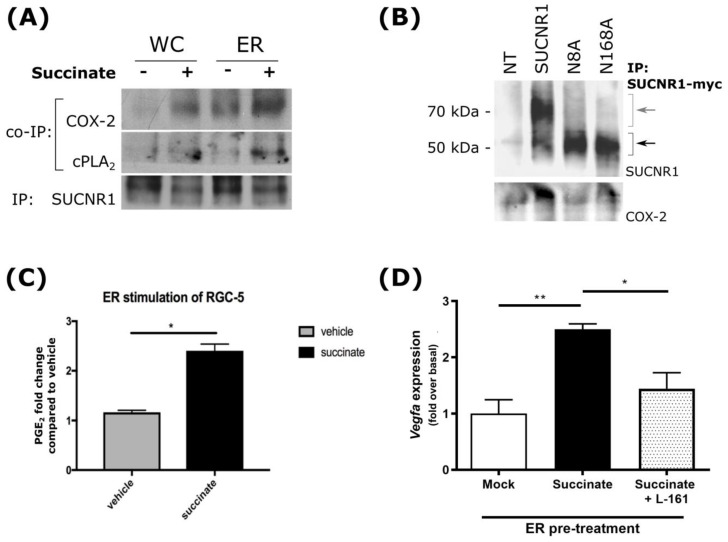
ER−localized SUCNR1 interacts with the PGE synthesis pathway. (**A**) Co-immunoprecipitation of cPLA_2_ or COX-2 with the endogenous SUCNR1 in whole-cell or ER fraction from RGC-5 treated (+) or not (−) with 100 μM succinate. Immunoprecipitation was performed with an antibody specific to SUCNR1, and is shown as IP loading control. COX-2 and cPLA_2_ coIP signals increased upon succinate stimulation. (**B**) Co-immunoprecipitation of COX-2 with SUCNR1-myc expressed in HEK 293T cells. IP was conducted with an anti-myc antibody, and the presence of SUCNR1 and COX-2 in precipitates was assessed using the corresponding antibodies. Signal from wild-type and mutant SUCNR1 are indicated by grey and black arrows, respectively. COX-2 signal was strongest in lysates from cells expressing the fully glycosylated, wild-type SUCNR1. (**C**) Succinate stimulation of isolated ER from RGC-5 led to the production of PGE_2_. N = 6 * *p* < 0.05. (**D**) RGC-5 ER fractions were treated or not with 100 μM succinate for 30 min at 37 °C. Subsequently, isolated nuclei from RGC-5 cells were incubated for 30 min at 37 °C with the corresponding ER fractions in the presence or absence of EP_4_ antagonist L-161,982, after which *Vegfa* expression was quantified by RT-qPCR. N = 4 (* *p* < 0.05, ** *p* < 0.01). (**C**,**D**) * and ** *p* < 0.05 and *p* < 0.01, respectively, vs. comparative values.

**Figure 6 cells-11-02185-f006:**
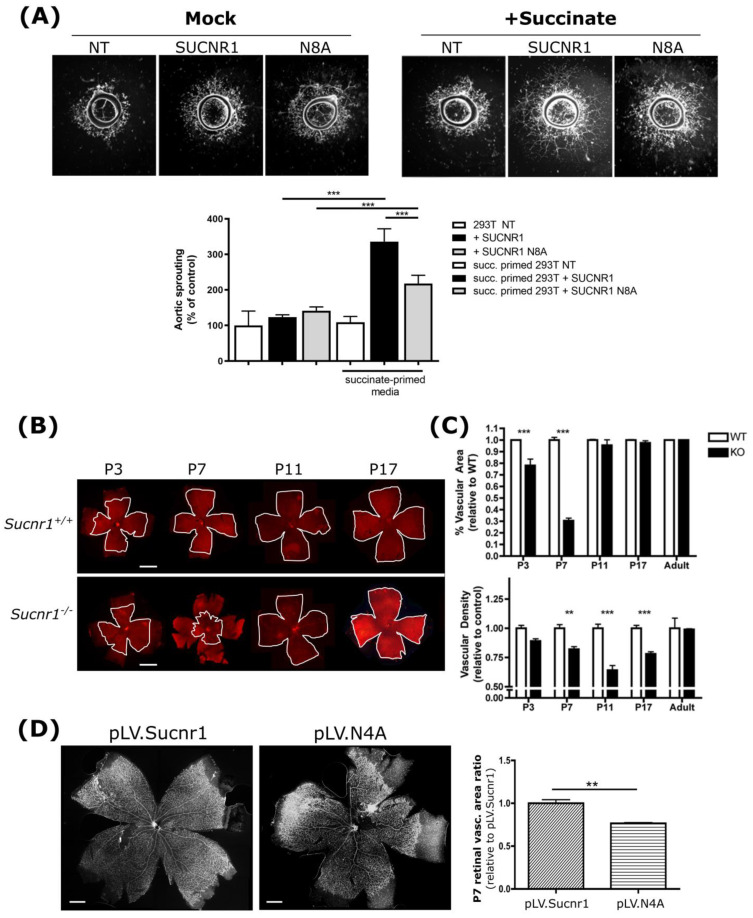
SUCNR1−N8A has a reduced retinal revascularization potential. (**A**) Female rat aortic explants were incubated in conditioned media from HEK 293T cells transfected with the corresponding SUCNR1 constructs, and stimulated or not with succinate (succ.). Aortic sprouting was quantified as the relative vascularized areas. Representative pictures (**left**) and corresponding histogram (**right**). n = 10; *** *p <* 0.001 vs. comparative values. (**B**) Representative images of lectin-stained flatmounts from *Sucnr1*^+/+^ or *Sucnr1*^−/−^ mice. Scale bars: 1 mm. (**C**) Quantification of images shown in (**B**). Retinal vascular surface area (**upper** panel) and vascular density (**lower** panel) in *Sucnr1*^+/+^ (white bars; wt) and *Sucnr1*^−/−^ (black bars) mice. P3 and P7 KO pups exhibited significantly reduced retinal vascular area and vascular density compared with wt pups. The reduced vascular density lasted until P17. ** *p <* 0.01, *** *p <* 0.001 compared with wt. (**D**) Representative pictures (**left** panel) and quantification (**right** panel) of the revascularized surface area in whole retinas following rescue of SUCNR1 expression in *Sucnr1*^−/−^ mice with a lentiviral plasmid overexpressing wt or N4A SUCNR1. The N4A mutant SUCNR1 showed reduced retinal revascularization (≈25%) compared with the wild type. The y-axis represents the ratio of retinal revascularization relative to the revascularization induced by the wild-type SUCNR1. Scale bars: 300 μm. Statistical analyses (n = 4): ** *p <* 0.01 vs. comparative values.

**Figure 7 cells-11-02185-f007:**
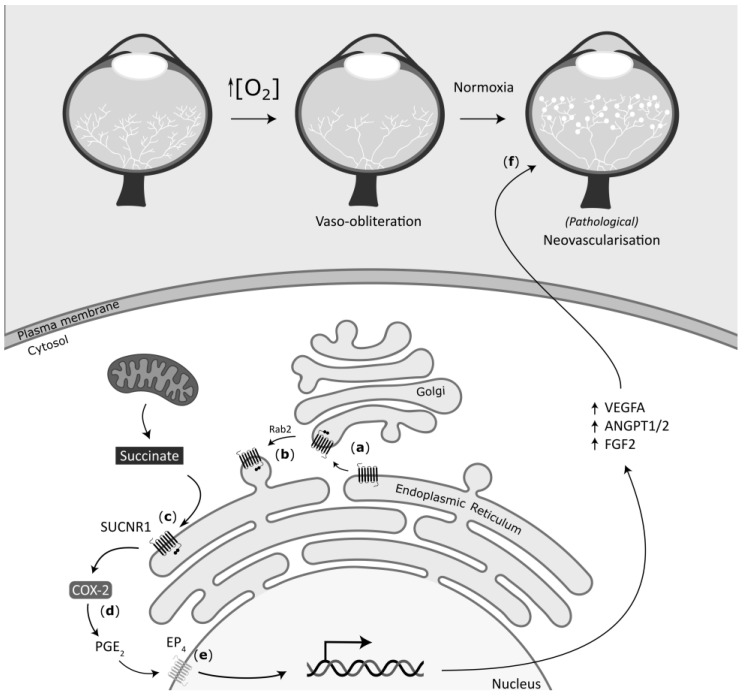
Schematic depiction of the angiogenesis mechanism induced by SUCNR1 stimulation. The top depicts the OIR model. When the retina is exposed to high O_2_ concentrations during development, a remarkable vaso-obliteration occurs, which, upon return to normoxic conditions, provokes local hypoxia. In this context, our proposed model of SUCNR1 signaling is: (**a**) SUCNR1 proper folding begins during translation at the ER, with glycosylation taking place along the ER-to-Golgi axis. (**b**) Upon proper glycosylation, RAB2-mediated SUCNR1 retrograde transport targets the receptor to the ER. (**c**) Upon increased energetic demands (hypoxia, intense exercise), succinate accumulates in the cytoplasm and stimulates SUCNR1. (**d**) SUCNR1 signals downstream through an interaction with COX-2, in turn leading to PGE_2_ production. (**e**) PGE_2_ stimulates the EP_4_ receptor, which subsequently triggers transcription of proangiogenic genes. (**f**) Higher proangiogenic gene expression leads to neovascularization; in prolonged hypoxic conditions where SUCNR1 localizes to the plasma membrane, disturbed expression of proangiogenic genes, notably lesser VEGFA expression, may be a factor driving the pathological “tufts” apparition.

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
