# Peer review of "The Succinate Receptor SUCNR1 Resides at the Endoplasmic Reticulum and Relocates to the Plasma Membrane in Hypoxic Conditions"

_cells, 2022, doi:10.3390/cells11142185_

Round 1

Reviewer 1 Report

This manuscript by Sanchez and colleagues investigates the function of the succinate receptor SUCNR1 in angiogenesis. Dr Chemtob’s laboratory has previously shown that SUCNR1 could induce pro-angiogenic and inflammatory responses in neurons by linking metabolic demand to vascular supply. Despite its importance in the regulation of these cellular processes, there is still little information regarding its downstream cellular events leading to its physiological role. In this study, they show that glycosylation of SUCNR1 regulates its cellular localisation in the endoplasmic reticulum, where it can alter its downstream signaling and expression of genes associated with the angiogenic response.

This study provides new understanding on the downstream signaling events regulating SUCNR1 function. The observation that SUCNR1 glycosylation regulates the localization and downstream effectors of the receptor is compelling. Overall, the data presented in the paper appears to support the proposed mechanism, but some of the data could be improved to further strengthen the conclusions of the manuscript.

Major

-          I don’t think that the figures shown in figure-4d support the conclusions that ‘’ In the intact mouse SUCNR1 IF signal (as seen in the ganglion cell layer) is primarily perinuclear (Fig 4d). However, during the hypoxic period of OIR, SUCNR1 staining diffuses throughout the cells and colocalizes with the PM-specific cadherin signal’’. Higher magnification pictures should be provided, as it is difficult to reach conclusions regarding the localisation of SUCNR1 staining for the provided images.

-          What is the biological significance of having PM- and ER-localized SUCNR1 triggering the expression of distinct pro-angiogenic genes (FGF2, Angpt for PM vs VEGF for ER)? This should be discussed. They claim that lower VEGF expression could be a factor for neovascular tuft formation, yet several studies show the importance of VEGF for neovascular tuft formation and that VEGF inhibitors can reduce tufts in OIR; is FGF-2 involved in tuft formation? This should be better addressed.

-          In the data presented in figure-6, it is stated that explant sprouting was not significantly increased by conditioned media from N8A mutant-expressing HEK293Tcells stimulated with succinate, and yet, they have also shown that succinate stimulation of N8A mutant-expressing cells lead to expression of several factors including FGF-2, which has potent pro-angiogenic properties. As such, why does supernatant from N8A mutant-expressing HEK293T cells not induce angiogenesis? This should be discussed.

-          In studies shown in figures 6A-C, controls using a VEGF neutralizing antibody should be used to confirm that the pro-angiogenic effects of the supernatant used indeed result from VEGF overexpression by succinate-treated HEK cells.

-          It is not obvious that there is increased vascular density in the eyes injected with succinate from the panels shown in Supp Fig. 3A. However, there seems to be an accumulation of microglial cells in the succinate condition, which could affect the quantification of vascular density. Was this considered when quantification was performed? The images should be replaced to show more representative pictures.

-          In experiments where lentiviral delivery of SUCNR1 constructs are used (fig. 6D), confirm expression by immunohistochemistry or immunoblotting. Also, figure legend for figure-6d is lacking.

Minor

-          Supp Fig 1a does not appear to be referenced in the manuscript

       In figure-3 and supplementary figure-1, show western blots and not only quantification

-          Fig6B-C: show representative images of retinas.

Reviewer 2 Report

Review Sanchez et al., 2022

SUCNR1 (or GPR91) has been increasingly recognized as a key regulator of numerous physiological functions in many different cell types. Therefore, new information about this receptor is an important and timely endeavor.  Towards that end, one of the most interesting points of this manuscript is the evidence suggesting that the SUCNR1 is primarily localized on ER membranes where it has a function.  Since that is the starting point of this manuscript, it needs to be rock solid. Unfortunately, there are caveats associated with much of the information in Fig. 1 purporting to show SUCNR1 on the ER. Fig. 1A and Supp Fig 1a is too small, fuzzy with uneven staining to convincingly show merged immunofluorescence. The most convincing data is the EM data in rat parietal cortex sections where anti-SUCNR1 immunogold particles primarily decorate outer ER membranes.  Because the EM data wasn’t quantitated, it’s difficult to know the relative amounts of receptor on the ER vs the plasma membrane although in the two lower power Images it would appear >50% of the receptor is on the ER. However as shown in Figure 1,B,II, there is certainly some SUNR1 at the plasma membrane which lessens the impact of the “functional” data shown in panels Fig. 1C-D. As is always the case with GPCRs, once activated even a very few receptors on the cell surface can lead to a large amplified signaling response. Without blocking cell surface receptors and/or isolating pure organelles and documenting their purity it’s difficult to draw conclusions on location specific SUCNR1-mediated Vegfa or Ca2+ expression. At the very least, using an ER-targeted Gcamp6 would have provided organelle-specific confirmation that activation of putative ER-SUCNR1 receptors led to a functional response (Neuron, 2017 93(4):867-881). Because these data are critical to the contention that SUCNR1 is a functional GPCR on the ER membranes, confirmatory evidence is necessary.

2) A second critical point that was not addressed here is how the intracellular receptor is getting activated. Given that the authors are applying succinate to whole cells and seeing effects, succinate, a polar, charged molecule, must be transported into the cell and across the ER membrane to elicit those effects. In Suppl. Fig. 3 authors do use probenecid to block succinate entry via an organic anion transporter (OAT). This might have been a useful tool in experiments 1C-F to determine what proportion of the SUCNR1 response was intracellular versus driven by the plasma membrane receptor.

3) Because there are many succinate transporters, the probenecid-sensitive OAT cell surface transporter may not be the same as the ER transporter. At the very least, a discussion of how succinate is accessing the ER lumen is also warranted.

4) Fig. 2D suffers from the same poor resolution as Fig. 1A. Given that these were transfected cells and thus over-expressed receptors; it should be possible to get clearer images more convincingly showing that the N8A mutant is now on the cell surface whereas the N168A is not.

5) Although glycosylation plays a role in targeting proteins to different subcellular locations, many other proteins binding to a variety of ER retention signals can also influence where a protein ends up. These should be discussed.

6) Fig 3 suggests that cell surface or E-localized SUNCR1 activates different signaling paradigms. Panels A and B should be statistically analyzed, Panel E seems to be highlighting the wrong comparison, bar should indicate a comparison between SUCNR1 with and without succinate treatment. All of these experiments would be stronger if 1) there was some proof of equal expression of transfected clones, 2) probenecid had been used to block the putative ER-mediated effects and 3) authors had tried to block the fgf2, Angpt1 and Anrpt2 effects using an ERK inhibitor since their results in Panels A and B ERK but not AKT was involved.

In general, the quality of many of the immunofluorescent panels is poor, the quality of the western blots is also marginal. Another major point throughout the entire manuscript is the lack of consistency and clarity. For example, Fig. 1C-F refers to Unt, which is not defined. In Sup. Fig. 5, it was finally defined as “untransfected”. That should be in the figure legends. However, “untransfected” is not an appropriate control! A vector-only transfected sample would have been much better. Fig. 3 then refers to “Mock” which presumably means “untransfected” but is never described either. Similarly, authors use a shNTC (non-targeting control) which is also sometimes referred to as shGFP and shSucnr1 is also referred to as ShGPR91. Suncr1 is sometimes lower case, other times uppercase. Some panels have been statistically evaluated whereas others haven’t.  There are also numerous miss-spellings and grammatical errors.

This is a very interesting and important manuscript, however, experiments haven’t been done in a clear or careful enough fashion to unequivocally draw the conclusions authors reach. Adding appropriate controls and inhibitors, improving figures, and evaluation, would significantly improve the manuscript.

Round 2

Reviewer 1 Report

While some clarifications were added in response to previous comments, some aspects of the manuscript still need improvements/clarifications:

The new statement in the discussion: ‘’ Sustained hypoxia relocates SUCNR1 to the cell surface which senses increased extracellular succinate to elicit vascular maturation factors, notably ANG1 and ANG2 and to a lesser extent FGF2.’’ I don’t necessarily agree that ANGPT2 and FGF2 could be described as maturation factors; in fact, ANGPT2 is widely known as a vascular destabilization agent that counteracts ANGPT1/TEK signaling. As such, the biological significance of having ANGPT1 and ANGPT2, which have opposing effects on TEK signaling, being upregulated at the same time is unclear. One could argue that having FGF2 and ANGPT2 upregulated together could destabilize vessels and promote a significant pro-angiogenic response. As such, the authors should be more careful about describing this response as an anti-angiogenic/maturation event, and expand on how these factors (ANGPT2, FGF2) might impact tuft formation.

While that I acknowledge that some of the structures shown in supplementary figure 3 could be diving vessels, the quality of the pictures shown is poor and blurry, and the retinas appear damaged which makes it hard to interpret; clearer micrographs should be presented.

Scale bars should be added to pictures.
